# Emerging Role of Oxidative Stress on EGFR and OGG1-BER Cross-Regulation: Implications in Thyroid Physiopathology

**DOI:** 10.3390/cells11050822

**Published:** 2022-02-26

**Authors:** Carmelo Moscatello, Maria Carmela Di Marcantonio, Luca Savino, Emira D’Amico, Giordano Spacco, Pasquale Simeone, Paola Lanuti, Raffaella Muraro, Gabriella Mincione, Roberto Cotellese, Gitana Maria Aceto

**Affiliations:** 1Department of Medical, Oral and Biotechnological Sciences, University “G. d’Annunzio” Chieti-Pescara, Via dei Vestini 31, 66100 Chieti, Italy; carmelo.moscatello@unich.it (C.M.); emira.damico@unich.it (E.D.); giordano.spacco@studenti.unich.it (G.S.); roberto.cotellese@unich.it (R.C.); 2Department of Innovative Technologies in Medicine & Dentistry, University “G. d’Annunzio”, Chieti-Pescara, Via dei Vestini 31, 66100 Chieti, Italy; dimarcantonio@unich.it (M.C.D.M.); luca.sav@hotmail.it (L.S.); raffaella.muraro@unich.it (R.M.); gabriella.mincione@unich.it (G.M.); 3Department of Medicine and Aging Sciences, University “G. d’Annunzio”, Chieti-Pescara, 66100 Chieti, Italy; pasquale.simeone@unich.it (P.S.); paola.lanuti@unich.it (P.L.); 4Center for Advanced Studies and Technology (C.A.S.T.) at University “G. d’Annunzio”, Chieti-Pescara, 66100 Chieti, Italy; 5Villa Serena Foundation for Research, 66013 Pescara, Italy

**Keywords:** oxidative stress, ErbB receptors, OGG1, MUTYH, BER, RET/PTC, PI3K/Akt, MAPK/ERK, thyroid, PTC

## Abstract

Thyroid diseases have a complex and multifactorial aetiology. Despite the numerous studies on the signals referable to the malignant transition, the molecular mechanisms concerning the role of oxidative stress remain elusive. Based on its strong oxidative power, H_2_O_2_ could be responsible for the high level of oxidative DNA damage observed in cancerous thyroid tissue and hyperactivation of mitogen-activated protein kinase (MAPK) and PI3K/Akt, which mediate ErbB signaling. Increased levels of 8-oxoG DNA adducts have been detected in the early stages of thyroid cancer. These DNA lesions are efficiently recognized and removed by the base excision repair (BER) pathway initiated by 8-oxoG glycosylase1 (OGG1). This study investigated the relationships between the EGFR and OGG1-BER pathways and their mutual regulation following oxidative stress stimulus by H_2_O_2_ in human thyrocytes. We clarified the modulation of ErbB receptors and their downstream pathways (PI3K/Akt and MAPK/ERK) under oxidative stress (from H_2_O_2_) at the level of gene and protein expression, according to the mechanism defined in a human non-pathological cell system, Nthy-ori 3-1. Later, on the basis of the results obtained by gene expression cluster analysis in normal cells, we assessed the dysregulation of the relationships in a model of papillary thyroid cancer with RET/PTC rearrangement (TPC-1). Our observations demonstrated that a H_2_O_2_ stress may induce a physiological cross-regulation between ErbB and OGG1-BER pathways in normal thyroid cells (while this is dysregulated in the TPC-1 cells). Gene expression data also delineated that *MUTYH* gene could play a physiological role in crosstalk between ErbB and BER pathways and this function is instead lost in cancer cells. Overall, our data on OGG1 protein expression suggest that it was physiologically regulated in response to oxidative modulation of ErbB, and that these might be dysregulated in the signaling pathway involving AKT in the progression of thyroid malignancies with RET/PTC rearrangements.

## 1. Introduction

Over the last 30 years, the incidence rate of thyroid cancer has steadily increased by 2.4 times [1,2] and papillary thyroid carcinoma (PTC) represents the most common histological type, with a frequency around 80% of cases [1]. Carcinogenesis and tumor progression in the thyroid gland are a phenotypic expression of a complex molecular interaction based on the connection between gene predisposition, environmental factors, and lifestyle, the effects of which influence thyroid hormone metabolism and oxidative DNA damage [3,4]. Oxidative damage has been suggested to promote tumor initiation and progression by increasing mutation rates and activating oncogenic pathways. Thyroid hormone synthesis is a complex and multistep physiological process that requires an adequate amount of H_2_O_2_ for oxidative iodination catalyzed by the enzyme thyroid peroxidase (TPO) [5,6,7,8]. On the other hand, based on its strong oxidative power, H_2_O_2_, could be responsible for the high level of oxidative DNA damage as observed in thyroid cancer tissue; it is also considered a second messenger able to activate several signaling pathways [9,10]. Oxidative DNA lesions have been detected in advanced stages of thyroid cancer, suggesting their contribution to tumor progression [4]. In recent years, there has been a growing interest in studying the role of oxidative stress in cancer initiation/progression and therapeutic response [11]. Indeed, H_2_O_2_ is also a signaling molecule involved in both the regulation of cell proliferation and apoptosis [10]. 

In cancer cells, H_2_O_2_ may regulate EGFR and mitogen-activated protein kinase (MAPK) signaling that contribute to redox protein-mediated cancer progression [12]. In addition, high levels of ROS have been found in thyroid adenomas and early stages of cellular transformation along with decreased antioxidant enzymes. Indeed, both oxidative stress and DNA damage are thought to be events that precede neoplasia in thyroid cells. [13] Such a circumstance, present in chronic thyroiditis, may contribute to the development of papillary thyroid carcinoma (PTC) [14]. H_2_O_2_ has been shown to cause the RET/PTC1 rearrangement frequently found in radiation-induced PTCs [15]. A marked increase in the presence of 8-oxoG in nuclear and mitochondrial DNA was evident after treatment with H_2_O_2_ or ionizing radiation [16]. Of the various types of oxidative DNA damage, 8-oxo-7,8-dihydroguanine (8-oxoG), has been reported as the most abundant. Despite its mild cytotoxicity, the 8-oxoG incorporation into DNA may be mutagenic since it causes a high transversion mutation rate [17,18,19]. Increased levels of the 8-oxoG DNA adducts have been detected in early stages of thyroid cancer: this may contribute to mutagenesis resulting in increased cell proliferation and survival [20]. The 8-oxoG DNA lesion is recognized and efficiently removed by 8-oxoG glycosylase1 (OGG1)-initiated base excision repair (BER) pathway (OGG1-BER) [20,21,22]. OGG1 glycosylase protects DNA from mutagenic potential caused by 8-oxoG:A mismatch leading to G:C to T:A transversion. Therefore, a correct function of OGG1-BER can be considered a protective factor against the triggering of carcinogenic pathways [20,21]. 

The involvement of EGFR signaling in thyroid carcinogenesis has been documented for a long time [23,24] and a link between DNA repair mechanisms and epidermal growth factor receptor (EGFR) signaling has been reported in many human tumor cells [25,26,27]; however, their cross-regulation is poorly understood in thyroid pathophysiology. 

All these observations prompted us to study the existence of cross-regulation between EGFR- and OGG1-BER-related molecular pathways in thyrocytes under H_2_O_2_-mediated acute oxidative stress. 

For our purpose, we compared a thyroid cancer (PTC) model with a normal cell counterpart to better identify changes in molecular signaling under specific treatment conditions, that would alter biological pathways for maintaining normal cell function. We chose to use a system of immortalized normal cells, Nty-ori 3-1, and a human model of differentiated papillary thyroid cancer (TPC- 1 cells). In the context of papillary carcinoma cell lines, the TPC-1 line has no mutations other than RET/TPC rearrangement [28]. 

This allowed us to elucidate how modulation of ErbB receptors and their downstream pathways (PI3K/Akt and MAPK/ERK) under acute oxidative stress are able to influence the molecular relationships with the OGG1-BER pathway.

## 2. Materials and Methods 

### 2.1. Cell Culture and Treatments 

Human thyroid follicular epithelial Nthy-ori 3-1 cells, obtained from European Collection of Authenticated Cell Cultures (ECACC 90011609) (Public Health England, Porton Down, Salisbury, UK) (Sigma Aldrich, St. Louis, MO, USA) were cultured at 37 °C in RPMI 1640 medium containing 10% fetal bovine serum (FBS), 100 U/mL penicillin/streptomycin and 2 mM L-glutamine. Human thyroid cancer cell lines, TPC-1 (harboring RET-PTC rearrangement, BRAF WT/WT), characterized according to Schweppe et al., and kindly provided by A. Coppa (Department of Experimental Medicine, Sapienza University of Rome, Rome), were maintained in a 5% CO_2_ culture humidified atmosphere, at 37 °C in Dulbecco’s modified Eagle medium (DMEM) supplemented with 10% FBS [29]. For experiments, cells were seeded and at sub-confluence (70%), were starved overnight using medium with 0.2% FBS, in order to reduce basal cellular activity [30,31]. Then, the cells were stressed by H_2_O_2_ (Sigma-Aldrich, Milan, Italy) alone or combined with EGF (human recombinant, Sigma Aldrich) or MAPK and AKT inhibitors (PD98059 and LY294002, respectively). In particular, the cells were pretreated with 50 ng/mL EGF for 15 min or with PD98059 (PD) at 50 μM and LY294002 (LY) at 25 μM (Cell Signaling Technology, Beverly, MA, USA) for 1 h and then stressed by a supra-physiological H_2_O_2_ concentration added to the external medium.

### 2.2. Cell Viability Assay 

To assess the H_2_O_2_ effects on cell viability, Nthy-ori 3-1 and TPC-1 cells were cultured in 96-well plate (1.0 × 10^4^ cells/well) for 48 h, starved overnight using medium with 0.2% FBS and then exposed to supra-physiological H_2_O_2_ increasing concentration (from 50 µM to 10 mM) at different times (3, 6 and 24 h). This was followed by incubation with 10 μL/well of 2-[2-methoxy-4-nitrophenyl]-3-[4-nitrophenyl]-5-[2,4-disulphophenyl]-2H-tetrazolium, monosodium salt (MTS) assay (Promega, Madison WI, USA) a 37 °C for 1 h. Cell viability was evaluated at 490 nm using the GloMax-Multi Detection System (Promega). The MTS assay is an indirect measure of cell number/vitality based on mitochondrial metabolic activity. The H_2_O_2_ stressor concentration range was chosen based on the physiological amounts of H_2_O_2_ reported as being produced in vitro by proliferating thyroid models [31].

### 2.3. Flow Cytometry and Cell Cycle Assay 

The effect of limit H_2_O_2_ stress on the quality of cell cycle at short time points was evaluated by flow cytometry cell cycle analyses (FACS) on normal and tumor cell lines. The cells were seeded on 6 cm plate at a density of 0.8 × 10^6^; at sub-confluence (70%), the cells were starved overnight (with 0.2% FBS), then were treated with H_2_O_2_ 10 mM for a short time (15 and 30 min). FACS analysis were carried out as already reported [32,33]. Briefly 3 × 10^5^ cells/sample were fixed by adding 500 μL of 70 % cold ethanol and then stored at 4 °C. After at least 24 h, samples were washed and stained by 500 μL of a solution composed of 50 μg/mL of propidium iodide (PI, Sigma) and 200 μg/mL of RNAse (Sigma). Cells were incubated overnight at 4 °C in the dark and then acquired by flow cytometry. PI fluorescence data were collected using linear amplification. Samples were acquired on a FACSCanto II flow cytometer (Becton Dickinson Biosciences, San Jose, CA, USA). Data were analyzed using the FlowJo software (Becton Dickinson Biosciences).

### 2.4. Real-Time Quantitative PCR Analysis (qRT-PCR)

For gene expression experiments, the cells were seeded on 10 cm plate at a density of 3.2 × 10^4^; at sub-confluence (70%), starved overnight using medium with 0.2% FBS, then the cells were treated with H_2_O_2_ alone or combined EGF or MAPK and AKT inhibitors (PD98059 and LY294002, respectively) and then acutely stressed for 15 and 30 min with a concentration of H_2_O_2_ (10 mM) determined on the basis of MTS and FACS assays.

Total RNA was isolated using TriFast (EUROGOLD EuroClone) according to the manufacturer’s instructions. The synthesis of complementary DNA (cDNA) was performed as previously described [34]. The mRNA levels were evaluated by SYBR Green quantitative real-time PCR (qRT-PCR) analysis using StepOne™ 2.0 (Applied Biosystems, Thermo Fisher Scientific, Waltham, MA, USA). Data were analyzed using the comparative Ct method and were graphically indicated as 2^−ΔΔCt^ + SD. In accordance with the method, the mRNA amounts of the target genes were normalized by the ratio on the median value of the endogenous housekeeping gene (*GUSB*) obtained in each treated cells vs. untreated (quiescent) cells.

Targets and reference genes were amplified in triplicate in a volume of 10 μL containing 1 μL template cDNA, 0.2 μL of primers mixture and 5 μL of GoTaq^®^ 2-Step RT-qPCR System (Promega) according to the manufacturer’s instructions. Primers sequences are available in Appendix A. The cycling conditions were performed as follows: 10 min at 95 °C and 40 cycles of 15 s at 95 °C, followed by 1 min at 60 °C, and final elongation of 15 s at 95 °C.

### 2.5. Gene Dosage Assay of OGG1 and MUTYH 

Two different genomic target sequences were selected for *OGG1* (GeneID: 4968; Gene Bank accession number: NM_016821) and *MUTYH* (GeneID: 4595; Gene Bank accession number: NM_12222.1). Genomic DNA was extracted as previously described [35] and gene dosage was performed by SYBR Green qRT-PCR. The samples were amplified in triplicate in three independent experiments. Data were analyzed using 2^−ΔΔCt^ + SD. In accordance with the qRT-PCR method, previously described, the gDNA amounts of the target genes were normalized by the ratio on median value of the *β-Actin* as genomic reference from TPC-1 vs. Nthy-ori 3-1. Primers sequences are listed in Appendix A.

### 2.6. Western Blotting 

For protein expression experiments, Nthy-ori 3-1 and TPC-1 cells were seeded on 10 cm plate at 3.2 × 10^4^ density, at sub-confluence the cells were starved overnight and then acutely stressed by 10 mM H_2_O_2_ for 15 and 30 min alone or combined with EGF or PD or LY. Total proteins were isolated from treated and untreated cells using lysis buffer [2 mM Na3VO4, 4 mM sodium pyrophosphate, 10 mM sodium fluoride, 50 mM HEPES pH 7.9, 100 mM NaCl, 10 mM EDTA, 1% Triton X-100, 2 µg/mL leupeptin, 2 µg/mL aprotinin, 1 mM PMSF]. Protein concentrations were determined using the BCA protein assay (Thermo Fisher Scientific, Waltham, CA, USA). An equal amount of total proteins was separated on 4–20% SDS-PAGE pre-cast gel electrophoresis (Bio-Rad Laboratories, Hercules, CA, USA) and transferred onto PVDF membranes (GE Healthcare, Chicago, IL, USA). Then, after blocking, the membranes were incubated with primary antibody overnight at 4 °C. The following primary antibodies were used: phospho-p44/42 MAPK, phospho-AKT, phospho-EGFR (Tyr1068), PARP-1 (Cell Signaling Technology); EGFR (Santa Cruz Biotechnology, Santa Cruz, CA, USA); ErbB2 (Dako, Santa Clara, CA, USA); OGG1 (Novus Biologicals, Littleton, CO, USA); β-actin (Sigma-Aldrich, St. Louis, MI, USA)—used as a protein loading control. Secondary antibodies were HRP-conjugated anti-rabbit or anti-mouse (Bethyl Laboratories, Montgomery, TX, USA). The immune complexes were visualized using the ECL Western blot detection system (EuroClone). Protein amounts were quantified by the Image Lab ^TM^ Software Version 5.0 (Bio-Rad Laboratories). 

### 2.7. Statistical Analysis and Tools 

All measurements were made after three independent experiments, and, for each, data is shown as a representative value of all experiments plus standard deviation. The results were subjected to t-test or one-way analysis of variance (ANOVA) as appropriate. All *p* values are two-sided and a *p* value of less than 0.05 was considered significant. All analyses were performed using SPSS software version 20 (IBM Corp., Armonk, NY, USA). The program Multiexperiment viewer v4.9.0 (MeV v4.9.0, J. Craig Venter Institute, La Jolla, CA, USA) [36] was employed to elucidate molecular relationship among genes expression detected in the normal Nthy-ori 3-1 cells in response to the treatments.

## 3. Results

### 3.1. Cell Line Viability after Hydrogen Peroxide (H_2_O_2_) Treatment

In starved cells after H_2_O_2_ treatment, we observed a rapid decline in cell viability, probably due to the interference of H_2_O_2_ with mitochondrial activity (*p* < 0.001) (Figure 1). In normal cells (Nthy-ori 3-1), viability settled at a constant value (approximately 50%) for all concentrations and times (Figure 1a). In contrast, tumor cells (TPC-1), after treatment, showed variability in their viability, at the different H_2_O_2_ concentrations used (Figure 1b). 

### 3.2. Cell Cycle Analysis

Based on the MTS results, we evaluated the acute oxidizing effect on the cell cycle quality of treatments with the highest concentration of 10 mM to test the acute effects of H_2_O_2_ at 15 and 30 min, in Nthy-ori-3-1 and TPC-1 cells. The percentages of cells in G0/G1, S and G2/M phases of the cell cycle were calculated using FlowJo software. FACS analysis on untreated Nthy-ori-3-1 underlined that 50.3 ± 6.1 % of cells were in G0/G1 phase, 24.8 ± 7.9 % in S phase and 25 ± 3.0 % in G2 phase in control group. The comparison with treated cells at 15 and 30 min resulted in a slight increase in the percentage of G0/G1-phase cells and in a reduction in G2 phase. H_2_O_2_ treatment seemed do not affect the S-phase cell population. Untreated TPC-1 cells compared with normal cells showed a higher percentage of G0/G1 phase cells (73.3% ± 3.5), a lower percentage of S phase cells (9.5% ± 3.9) and the remaining 17.2 ± 5.2 in G2. Treated TPC-1 cells showed a reduction in G0/G1 and an increase in G2 phase cell percentage (Figure 2a,b).

### 3.3. Gene Expression of BER and EGF Signaling in Nthy-ori 3-1 vs. TPC-1

Gene expression modulation of BER and EGF signaling was assayed in starved human thyroid follicular epithelial Nthy-ori 3-1 cells and starved papillary carcinoma cell line TPC-1 after short term treatments (15 and 30 min), with H_2_O_2_ [10 mM], EGF, PD, LY alone or combined. 

#### 3.3.1. OGG1-BER Signaling

The BER genes—*OGG1, MUTYH, APE-1/Ref-1* and *PPARγ*—in Nthy-ori 3-1 exhibited generally the same trend in the different experimental conditions (Figure 3a). These genes showed a significant (*p* < 0.05) time-dependent increase, after H_2_O_2_ exposure, and a significant (*p* < 0.05) decrease after EGF stimulation and AKT and MAPK inhibition (Figure 3a). EGF and H_2_O_2_ co-treatment increased *OGG1*, *MUTYH* and *PPARγ* compared with EGF stimulation alone; this was not observed for *APE-1*. With respect to normal cells, the TPC-1 tumor cells showed an overall gene expression deregulation, in response to the treatments (Figure 3b vs. Figure 3a). Indeed, an increase in expression after AKT inhibition was observed for *OGG1* and *APE1*, while *PPARγ* and *APE1* showed an overexpression after stimulation by EGF (Figure 2b). *MUTYH* expression was not appreciably detected in TPC-1 cells (data not shown).

#### 3.3.2. EGF Signaling

In Nthy-ori 3-1 cells, the stimulation with EGF and H_2_O_2_, alone or combined, did not induce any *EGFR* modulation; while LY and PD determined its overexpression and furthermore H_2_O_2_ strengthened the action of inhibitors (Figure 4a). ErbB2 expression was increased by H_2_O_2_ combined with EGF and MAPK inhibitor. ErbB2 expression was increased by co-treatment with H_2_O_2_ and EGF, while it was downregulated by EGF alone and by AKT inhibitor. (Figure 4a). *ErbB3* was stimulated by EGF and PD treatments, also combined with H_2_O_2_ (Figure 4a). In all conditions *ErbB4* was not detected, except after EGF stimulation (data not shown). An aberrant modulation of the *EGFR*, *ERBB2* and *ERBB3* genes was observed in TPC-1 cells. In fact, these genes tended to be repressed or poorly expressed (even in proliferating cells) but they resulted stimulated by MAPK inhibition combined with H_2_O_2_ (Figure 4b).

#### 3.3.3. Oxidative Stress Signaling

The gene expression levels of the molecules known to be involved in the oxidative stress response are shown in Figure 4.

In Nthy-ori 3-1, the nuclear factor erythroid-derived 2-like 2, *(NRF2),* decreased with EGF, LY and PD (alone and combined with H_2_O_2_) (Figure 5a). The inhibition by PD and PD + H_2_O_2_ after 30 min increased the expression of protective antioxidant gene *HO1*, that conversely was decreased by EGF and LY inhibitor (Figure 5a). In cancer cells, *NRF2* showed a significant alteration in its expression, especially after treatment with H_2_O_2_, but also with LY and PD (Figure 5b).

In normal cells, the transcription factor *JUN/AP1* did not show any modulation following H_2_O_2_, LY and PD treatments, but its expression was increased by EGF (Figure 5a). In tumor cells, EGF stimulation resulted in overexpression of JUN/AP1 while inhibition of AKT downregulated it (Figure 5b). The pro-inflammatory CXC chemokine *IL-8* gene (*CXCL8*) was significantly increased in normal proliferating cells compared with normal starved cells and in particular after H_2_O_2_ treatments, whereas its expression was reduced by EGF stimulation and its downstream pathways (AKT and MAPK) inhibition (Figure 5a). In cancer cells, *IL-8* expression is dramatically reduced by MAPK inhibition.

#### 3.3.4. Stemness and Differentiation Markers

As expected, in Nthy-ori 3-1, the expression of stem cell regulator *ZEB1* increased in proliferating cells, after EGF stimulation and in stress conditions, while its expression was decreased by LY and H_2_O_2_ co-treatment (Figure 6a). Thyroid differentiation marker *TPO* showed an opposite trend in gene expression compared with *ZEB1*, except for the treatment with H_2_O_2_ (Figure 6a). The reduction in *TPO* gene expression was detected following the inhibition of the AKT pathway while an opposite effect was obtained after MAPK inhibition (Figure 6a). In TPC-1 the expression of zinc finger *ZEB1* increased after LY and PD treatments, while H_2_O_2_ reduced its expression (Figure 6b). In all experimental conditions, thyroid differentiation marker *TPO* resulted undetected (data not shown).

### 3.4. Gene Dosage Assay of OGG1 and MUTYH in TPC1 Cells

Since TPC-1 cells did not express *MUTYH*, we evaluated the amount of genomic DNA (gDNA) of this gene and its partner *OGG1.* The qPCR data were normalized by the ratio on the value of endogenous *B-Actin* gDNA. The reference amount of the both alleles presence was validated on gDNA from Nthy-ori 3-1 cells; since the normalized values of the controls (cell lines) matched perfectly, we were confident to employ this method to assay the TPC-1 cell line genome. In the TPC-1 cells, *OGG1* and *MUTYH* gDNA values did not show reduction (Figure 7).

### 3.5. Gene Expression Cluster Analysis in Nthy-ori 3-1

To elucidate the potential cross-regulation between BER and EGFR pathways, we compared the genes expression levels detected in Nthy-ori 3-1 cells in response to the several treatments employing Multiexperiment viewer v4.0 program (MeV4.0) [35] (Figure 8). Under acute oxidative stress conditions (by 10 mM H_2_O_2_ treatment), a close correlation among genes involved in stress response, i.e., *MUTYH, HO1* and *OGG1,* was observed. We also highlighted the correlation among TPO, IL-*8* and *JUN/AP1* transcription factor (Figure 8a).

When the Nthy-ori 3-1 cell line was stimulated by EGF, a strong relationship among the BER genes, *OGG1 and APE1/Ref1 (alternatively named APEX1, HAP1, APEN),* and antioxidant response gene, *NRF2* and *OH1* was shown; whereas *MUTYH* and *PPAR*γ showed a correlation with *ErbB2* (Figure 8b). The inhibition of AKT pathway by LY, highlighted a lack of BER gene correlation, in addition to *APE1/Ref1,* was related to *NRF2*, *HO1 IL-8* and *ErbB2* (Figure 8c). Interestingly, in normal thyroid cells the inhibition of MAPK (by PD) was the unique experimental condition that showed a correlation among the ErbB receptors (Figure 8d). 

### 3.6. Protein Expression of ErbB Pathway and OGG1 in NThy-ori 3-1 and TPC-1 Cells

The effect induced by acute H_2_O_2_ stress on ErbB and OGG1 pathways was also assessed on protein expression. Western blotting analyses showed that the OGG1 protein was expressed in quiescent Nthy-ori 3-1 cells, with clearly visible bands, at approximately 50 kDa (Figure 9a) [37,38]. This OGG1 isoform was reduced or absent after H_2_O_2_ treatments alone or combined with AKT and MAPK inhibitors (LY and PD, respectively). Instead, the same bands increased after PD and EGF treatment (Figure 9a), while EGF and H_2_O_2_ co-treatment seemed to induce further OGG1 post-transcriptional modifications, thus allowing the detection of 37kD isoform (Figure 9a).

The 37 kDa band, which should correspond to the nuclear OGG1 isoform (1a) [37] was increased by EGF and H_2_O_2_ co-treatment. The level of PARP-1 protein expression was very mild in quiescent NThy-ori 3-1 and TPC-1 cells and the oxidative stress did not seem to modulate it (data no shown). Under all the experimental conditions we used, this protein resulted not cleaved, indicating that apoptosis (PARP-1-dependent) has not been activated. 

Unlike what observed in normal Nthy-ori 3-1, in TPC-1 cells H_2_O_2_ increased OGG1 expression after MAPK pathway inhibition, and EGF stimulation caused relevant changes respect to untreated quiescent cells (Figure 9a vs. Figure 9b). In Nthy-ori 3-1 cell line, the protein level of EGFR remained unchanged upon all treatments (data not shown), while, as expected, its phosphorylation in Tyr1068 was absent in starved untraded cells and appeared after EGF stimulation (Figure 9a). The acute stress by H_2_O_2_ always induced EGFR phosphorylation even after inhibition of the AKT and MAPK. In all experimental conditions, ErbB2 was expressed and slightly reduced following oxidative stress (Figure 9a). In quiescent cells, MAPK phosphorylation was observed, while it was lightly inhibited by treatment with LY alone and with EGF plus H_2_O_2_. PD and H_2_O_2_ co-treatment increased MAPK phosphorylation compared with PD alone. In quiescent untreated cells, a high level of phospho-AKT was detected, and H_2_O_2_ treatment, after 30’, inhibited it so as to be undetected. PD treatment led a decrease in AKT phosphorylation which was more pronounced after co-treatment with H_2_O_2_. 

Papillary thyroid cancer model TPC-1 did not express p-EGFR under any of the tested conditions (data not shown), whereas ErbB2 protein was always expressed and particularly increased after PD and EGF alone treatments (Figure 9b). The MAPK were activated in all experimental conditions especially with LY and H_2_O_2_ co-treatment, contrary to what observed in normal cells. AKT was activated in the presence of H_2_O_2_ also with PD and EGF alone. Relative expression quantification of the analyzed proteins in Nthy-ori3-1 and TPC-1 cells are visible in Figure 10.

## 4. Discussion

Thyroid diseases have a complex and multifactorial etiology [39]. Despite the numerous studies on the signals referable to the malignant transition [3], the molecular mechanisms concerning the role of oxidative stress in diseases and carcinogenesis of thyroid remain elusive. Indeed, it has long been believed that oxidative stress plays an active role in carcinogenesis and cancer progression [5]. One of the molecules with pro-oxidant characteristics is H_2_O_2_. It is physiologically produced and is involved in the adaptation to stress and in chronic inflammatory responses [9,10], but it is also actively produced in the thyroid as necessary for thyroid hormone synthesis. However, there are still many gaps in our knowledge regarding the cellular signaling network attributable to H_2_O_2_ and to DNA damage in thyroid. Based on its strong oxidative power, H_2_O_2_ in cancer cells with high proliferative rate could be responsible for the high level of oxidative DNA damage as observed in cancerous thyroid tissue [15] and hyperactivation of some signaling pathways [10] including mitogen-activated protein kinase (MAPK), and PI3K/Akt, involved in ErbB signaling [13,40]. 

Several studies on exposure of thyroid cells to irradiation or to nonlethal concentrations of H_2_O_2_ have shown that H_2_O_2_ can cause DNA damage, also supporting the hypothesis that H_2_O_2_ generation in the thyroid could play a role in mutagenesis, especially when antioxidant defense is deficient. [4]. Indeed, marked increases in the presence of 8-oxoG in nuclear and mitochondrial DNA were evident after treatment with H_2_O_2_ or ionizing radiation. [16]. In aqueous solutions, as in the cellular interior, the reaction of H_2_O_2_ with guanine leads to the formation of 8-oxoG complexed with a water molecule can occur according to two different mechanisms that have two steps each. The processes are described in detail by Jena et colleagues [41]. H_2_O_2_ exposure has also been widely used to study the dynamics of 8-oxoG in mitochondrial DNA in wild-type and OGG1-deficent cell models. [42] Wang and colleagues used H_2_O_2_ to simulate oxidative stress. They stated that although it is not a free radical, H_2_O_2_ can circulate in cells and be efficiently converted to a hydroxyl radical and lead to an overall increase in genomic 8-oxoG level even in cells with wild-type OGG1. This increase in DNA lesions in proliferant cells was detected starting at 5 min of treatment, peaking at 15 min, and were then rapidly repaired by wild-type OGG1, restoring the redox balance. The authors also argued that the high level of 8-oxoG itself might not increase cell death [22].

In this study, we investigated the relationship between the EGFR and OGG1-BER pathways and its regulation following acute oxidative stress by H_2_O_2_ in quiescent human thyroid cells. We evaluated, in both starved cell models, the effects of H_2_O_2_ on cell viability. Initially, the range of concentrations chosen was supposed to mimic momentary distress compared with physiological amounts of H_2_O_2_ produced by thyroid models in vitro [33], but also to what might be produced in the tissue microenvironment, e.g., under infectious and/or inflammatory conditions [10]. 

The H_2_O_2_ response of cultured cells appears to depend on many factors [41], In starved cells, we observed a rapid decline in cell viability after H_2_O_2_ treatment due to H_2_O_2_ interference with mitochondrial activity (Figure 1) and apoptosis/death (Appendix A) [43]. In additional analysis of cell cycle, after H_2_O_2_ treatment, we did not observe sub-G1 cells in the H_2_O_2_-treated samples (Figure 2a). TPC-1 tumor cells showed a recovery of viability at 3 h with the lowest concentrations of H_2_O_2_, this phenomenon also appears to occur in other tumor cell models [44] and could be due to the multiple functional ormetic properties of H_2_O_2_ in the cell context [44,45,46]. On the other hand, we must also consider that in thyrocytes the TPO downregulation, reducing its H_2_O_2_ chemo-protective function, is involved in the evolution of nodularity towards the condition of tumorigenesis [47], in TPC-1 the expression of TPO was not detected. H_2_O_2_, is a crucial substrate for thyroid peroxidase (TPO), a key enzyme involved in thyroid hormone synthesis [8]. On the plasma membrane of thyrocytes, TPO is associated with oxidoreductase dual oxidase 2 (DUOX2) and this functional interaction is essential for the regulation of the extracellular H_2_O_2_ level [8]. 

Based on the above considerations, in this study we highlighted the crosstalk effect between EGFR and OGG1-BER on gene expression modulation, by stimulating starved thyroid cells with H_2_O_2_ and EGF, and inhibiting EGFR its downstream ways, AKT and MAPK, using LY and PD (respectively) for a short time (15’ and 30’). We assessed the physiological relationships between the two pathways using a normal thyroid cell line (Nthy-ori 3-1), and verified the dysregulations of these signals in TPC-1 tumor cells. Nthy-ori 3-1 cells were treated with H_2_O_2_, EGF, LY and PD alone and in combination. We observed that expression of genes involved in BER pathway, i.e., *OGG1, MUTYH, APE-1/Ref1* and *PPARγ*, exhibited generally the same trend, resulting upregulated by H_2_O_2_ and downregulated by LY and PD. The *NRF2* gene expression decreased with EGF, LY and PD (with and without H_2_O_2_), while in contrast *OGG1* increased. This suggested that in thyroid *OGG1* may be regulated by a NRF2-indipendent way, in spite of the presence of an ARE (antioxidant response elements) region recognized by NRF2 in *OGG1* promoter [48]. In addition, the expression of the *HO1* gene, which responds both to oxidative stress and to NRF2-mediated gene regulation in different tissues [49], displayed a different behavior with respect to *NRF2*. These data did not completely exclude a role of NRF2 transcription factor in the response to acute oxidative stress in our system, because it is well-known that there are inactive form resides as subcellular compartmentalization pool bounded to Keap1 [50]. Gene expression cluster analysis showed that when Nthy-ori 3-1 cells were stimulated with EGF and H_2_O_2_, *OGG1* and *APE1/Ref1* showed a strong correlation with *NRF2* and its downstream *HO1* and *IL8*. Instead, *MUTYH* demonstrated a correlation with *ErbB2* and *PPARg* (Figure 8a,b). After LY treatment no significant correlation between any of the BER genes was shown (Figure 8c). Interestingly, MAPK inhibition (with PD) was the only experimental condition in which we observed a correlation between ErbB genes, this suggests a positive feedback mechanism in control gene expression (Figure 8d). Our results highlighted that the inhibition of ErbB receptors downstream pathways may have a role in dysregulation of the BER genes.

After inhibition of the AKT pathway, a correlation was observed between *OGG1* and *JUN/AP1* while after MAPK inhibition (Figure 8c,d), they correlated with *CXC* chemokine *IL-8* (also *CXCL8*) and *JUN/AP1*, as reported in the literature [51]. Therefore, the JUN/AP1transcriptional activator could play a pivotal role in the correlation between the ErbB and BER systems (Figure 7).

Pro-inflammatory CXC chemokine IL-8 (*CXCL8*) gene expression was significantly increased in Nthy-ori 3-1 proliferating cells and after H_2_O_2_ treatments compared with starved cells. On the contrary, *IL-8* was reduced after EGF, LY and PD treatments alone and combined with H_2_O_2_ (Figure 4). Our data confirmed that IL-8 plays a role in response to oxidative stress that also stimulate its own production [52]. Interestingly, its synthesis may be dependent by EGFR activation [53]. IL-8 was the first demonstrated chemokine to be secreted by human normal thyrocytes [54]. Moreover, the secretion of IL-8 by cancer cells can enhance the proliferation and survival capabilities through an autocrine loop [53]. *MUTYH* gene in Nthy-ori 3-1 cells was clustered with other BER genes under oxidative stress conditions, whereas EGF induced a correlation with *PPARγ* and *ErbB2* (Figure 2 and Figure 3), but this correlation was lost after LY and PD treatment suggesting that *MUTYH* gene expression might be regulated by EGFR pathways. Furthermore, gene expression data underlined a close correlation among *MUTYH, HO1* and *OGG1*, following H_2_O_2_ treatment. In contrast, following EGF treatment, a close correlation among *MUTYH*, *PPARγ* and *ErbB2* was shown. These results confirmed the relationship between ErbB2 and PPARγ in regulating cell proliferation and metabolism as already displayed in other tissues [55]. It also highlights a bivalent role for *MUTYH*, under acute oxidative stress condition *MUTYH* showed a correlation with the other BER genes, probably because the primary necessity of cells is to defends itself from the high level of oxidative stress, whereas after EGF growth stimulus *MUTYH* showed a correlation with ErbB2 and PPARγ likely due to an increase in the metabolism rather than DNA repair, which becomes a secondary mechanism.

In Nthy-ori 3-1 cells, EGF treatment increased *ErbB2* gene expression only in combination with H_2_O_2_. The same effect was observed by blocking MAPK pathway by PD (Figure 3a). Under the effect of LY and PD, cotreated with H_2_O_2_, EGFR expression increased significantly (Figure 3a). These data were confirmed by protein analysis showing changes in EGFR receptor activation after H_2_O_2_ treatment without altering its expression (Figure 9). These results led us to suggest the presence of fine-tuned mechanisms that regulate signals downstream of EGF and crosstalk with activities controlled by OGG1 protein. The simultaneous presence of a factor that stimulates cellular growth (i.e., EGF) and adaptation to the H_2_O_2_-mediated oxidative stress, induced the expression of different isoforms of the OGG1 in these cells. Our results showed that OGG1 protein in normal thyroid cells was expressed in several isoforms depending on the treatments used (Figure 8). These findings could be explained both on the basis of that is currently known about this enzyme; an alternative splicing mechanism that acts on the C-terminal region was delineated. As a consequence, all isoforms share the N-terminal region but differ in C-terminal, determining the existence of several isoforms, such as the alternative splicing process of the C-terminal region of OGG1 [35,36] and on post-translational modifications occurred as a response to the stress [56]. Currently, most of the activities of these isoforms are unknown. 

In the normal system, OGG1 was shown to be controlled by the AKT pathway as it also decreased in protein level, while the stimulation with EGF increased its expression in many of its isoforms (Figure 9). Our results highlighted that the inhibition of ErbB receptors downstream pathways may have a role in modulation of the OGG1 end BER. This aspect is very dysregulated in the TPC-1 tumor cell which shows a strong expression of OGG1 despite the inhibition of AKT (Figure 9b). 

In Nthy-ori cells, upon H_2_O_2_ treatment, AKT activity decreased under all conditions tested, whereas in TPC-1 cells, AKT activity increased upon H_2_O_2_ treatment. This difference in signaling between the two models could be explained by H_2_O_2_-driven deregulation in the mTOR pathways in TPC tumor model. H_2_O_2_, acting as an intracellular messenger, can activate phosphatidylinositol-3-kinase (PI3K) and its downstream target Akt, promoting cell survival. Accumulation of ROS has been associated with activation/phosphorylation of AKT to inactivation/phosphorylation of tuberin and activation of mTOR/complex II. Activation of adenosine-monophosphate-activated protein kinase (AMPK) by rapamycin may directly impact OGG1 activation or indirectly through blocking mTOR activity. This unusual activation of AKT is also followed by a higher protein level of OGG1 in the LY + H_2_O_2_ condition, which is not reflected in mRNA levels. The increase in OGG1 was detected in several tumor cells treated to suppress mTOR activity. Rapamycin treatments can inhibit mTORC1 through increased phosphorylation of AMPK and lead to increased OGG1 activity. Although lower concentrations of rapamycin appeared to inhibit the activation of mTOR activation in cancer cells, higher concentrations were required to increase the protein and promoter activity of OGG1, suggesting that high concentrations of rapamycin may be required to activate other kinases, such as AMPK, in addition to blocking mTOR activity [57]. In support of this hypothesis, rapamycin increases the phosphorylation of AMPK, a signaling intermediate upstream of the mTOR pathway in several tumor cells. Activation of AMPK by rapamycin results in the inhibition of mTOR activity and increased protein expression and promoter activity of *OGG1*. In addition, activation of AMPK, by 5-aminoimidazole-4-carboxamide riboside (AICAR), resulted in inhibition of mTOR activity and increased promoter activity and protein expression of OGG1 in tuberin-deficient and von Hippel–Lindau (VHL) deficient cells [57].

The mechanism of regulation of cellular OGG1 protein, particularly in response to oxidative stress, is unclear. The stability of OGG1 is increased in response to oxidative stress induced by ionizing radiation. As a consequence of prolonged OGG1 stability and increased excision activity in the absence of E3 ubiquitin-protein ligase NEDD4-like, cells show increased DNA repair capacity but, conversely, this decreases post-irradiation cell survival. This effect may be reproduced upon overexpression of OGG1, suggesting that dysregulation of OGG1 increases intermediate DNA lesion formation [58]. Therefore, it would be very interesting to further investigate these functional aspects of OGG1 in papillary thyroid carcinoma with RET/TPC rearrangements.

Although OGG1-BER protects the genome integrity by repairing the lesion or eliminating cells with malignant potential and its overexpression improves H_2_O_2_-induced cell death [22], in the case of TPC-1 cells, the lack of *MUTYH* gene expression could favor the maintenance of tumor phenotype loop [59]. Furthermore, this confirms that OGG1, as a sensor in the BER response, can be controlled both at the gene and protein level by the Pi3K/AKT pathway, while this was not detected following stimulation by EGF and H_2_O_2_. Probably in quiescent TPC-1 cells, the presence of the RET/PTC rearrangement keeps the growth pathways independent of EGF stimulation by not expressing p-EGFR, as it appears to be (albeit slightly) in TPC-1 proliferating cells [60]. Molecular studies have also shown that RET/PTC in human thyrocytes promotes the activation of inflammation-related genes expression, and this could contribute to the progression and locoregional metastases, characteristic of the PTC tumors [60,61]. Stimulation of the OGG1 isoforms could also be caused by this aspect. Indeed, some studies have shown that the 8-oxoguanin glycosylase-1 repair enzyme is also involved in inflammation regulation and diseases [62,63]. 

TPC-1 cells showed a strong deregulation of the BER, OGG1 and MUTYH, components (Figure 2) with a loss in the gene expression of *MUTYH* and an increased expression of *APE-1/Ref1*. This result could be due to the presence of the RET/PTC rearrangement in the papillary tumor model; indeed, oncogenic potential of RET/PTC is related to intrinsic tyrosine kinase activity and repression of p53-dependent transactivation [64]. Moreover, some previous studies have shown that the gene expression of *MUTYH* is controlled by TP53 [65] and that TP53 can also modulate the transcriptional activity of APE-1/Ref1 [66]. At present, these interactions between tyrosine kinase activities and BER signaling modulations have not yet been adequately explored in thyroid carcinogenesis and deserve further study. 

## 5. Conclusions

Our observations highlighted that human thyroid cells, under acute oxidative stress, activate a physiological cross-regulation between the ErbB and OGG1-BER pathways. This regulation, assessed in a normal thyroid epithelial cell model and in a differentiated papillary cancer model, appeared to be dependent on H_2_O_2_- and EGF-related oxidative signals. This was evidenced by the significant positive response of OGG1 protein to inhibition of MAPK and AKT signaling, as well as the complete inhibition of *MUTYH* gene expression and the very strong increase in *APE1* gene expression. Our data on OGG1 protein expression suggest that it was physiologically regulated in response to oxidative modulation of ErbB, and that these might be dysregulated in the signaling pathway involving AKT in the progression of thyroid malignancies with RET/PTC rearrangements. Overall, this study suggests further investigation of the pathophysiological interactions of OGG1-BER pathways to elucidate the mechanisms underlying the onset and development of thyroid diseases. 

## Figures and Tables

**Figure 1 cells-11-00822-f001:**
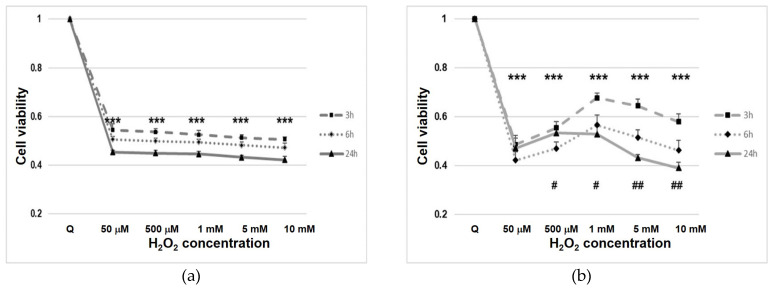
Nthy-ori 3-1 (**a**) and TPC-1 (**b**) cell line viability under H_2_O_2_ stress condition. Cell viability was assayed using MTS. The cells were exposed to H_2_O_2_ for different time and concentrations (from 50 µM to 10 mM). For each experiment, n = 5 replicates wells were assayed per clone. Cell viability values were calculated as means and compared with untreated quiescent cells (Q). ***—*p* < 0.001 vs. quiescent; #—*p* < 0.05; ##—*p* < 0.01 cells with same dosage of H_2_O_2_ treatment at different time.

**Figure 2 cells-11-00822-f002:**
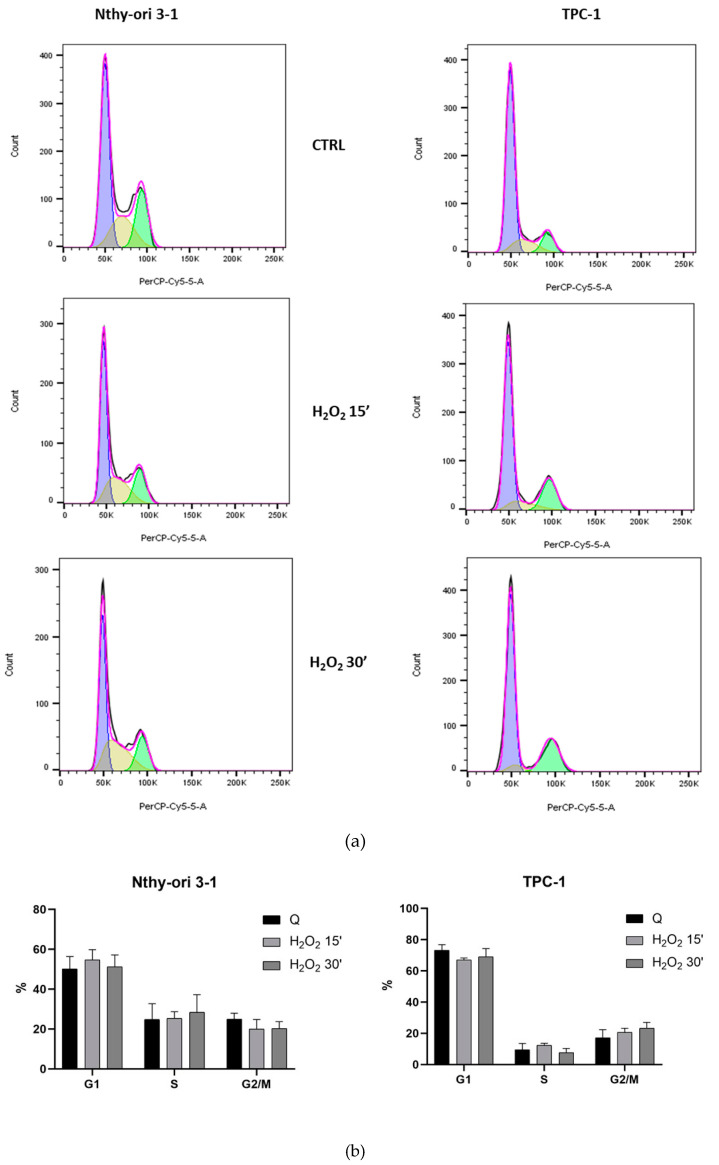
Effect of H_2_O_2_ [10 mM] on the cell cycle of starved thyroid cell lines. (**a**) FACS analysis profile of Nthy-ori 3-1 and TPC-1 untreated and treated cells by 10 mM H_2_O_2_ after 15 or 30 min; (**b**) relative percentages in cell cycle stages.

**Figure 3 cells-11-00822-f003:**
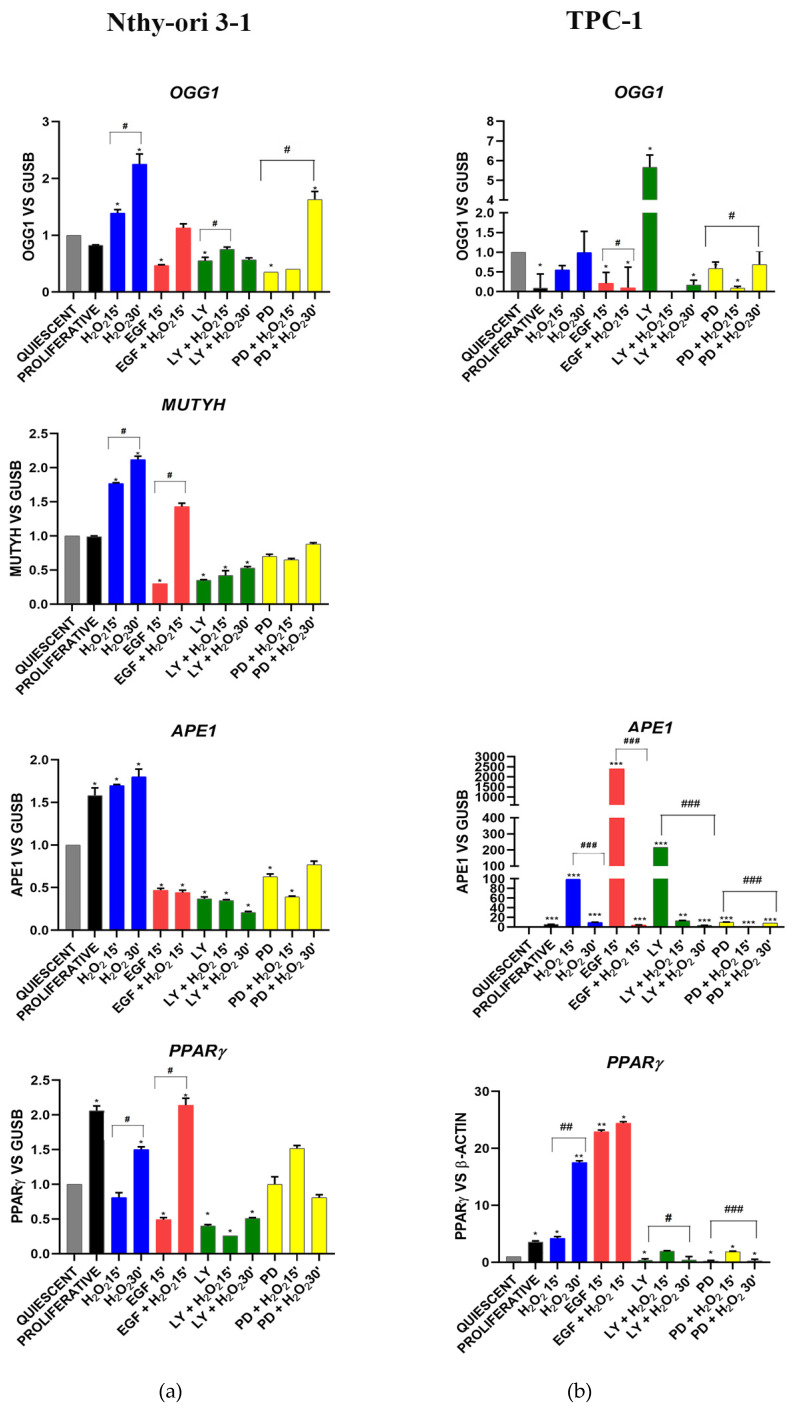
*OGG1* (8-Oxoguanine glycosylase), *MUTYH* (mutY DNA glycosylase), *APE1* (apurinic/apyrimidinic endodeoxyribonuclease-1) and *PPARγ* (peroxisome-proliferator-activated receptor gamma) gene expression modulation under H_2_O_2_ [10 mM], EGF, LY and PD treatments alone and combined in Nthy-ori 3-1 cells (**a**) vs. TPC-1 cells (**b**). Gene expression was analyzed by real-time qPCR. The histogram represented normalized data with *GUSB* gene in Nthy-ori 3-1, while the *PPARγ* expression data were normalized with β-Actin in TPC-1 cells. The expression of *MUTYH* was undetected in TPC-1 cells (five independent experiments). The results showed the average of three independent experiments. LY—LY294002; PD—PD98059; *—*p* < 0.05; **—*p* < 0.01; ***—*p* < 0.001 treated vs. quiescent cells; #—*p* < 0.05; ##—*p* < 0.01; ###—*p* < 0.001 cells with similar treatment.

**Figure 4 cells-11-00822-f004:**
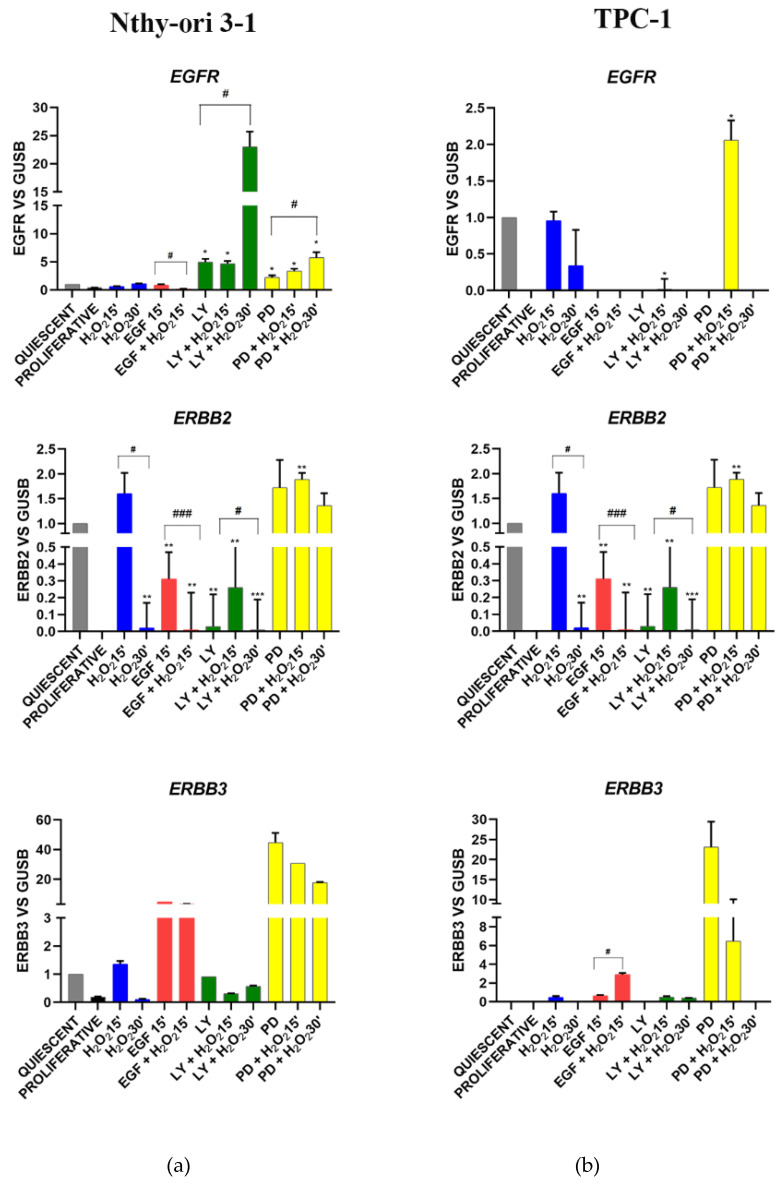
*EGFR* (epidermal growth factor receptor, *ERBB2* (Erb-B2 receptor tyrosine kinase 2), *ERBB3* (Erb-B3 receptor tyrosine kinase 3) gene expression modulation under H_2_O_2_, EGF, LY and PD treatments alone and combined in Nthy-ori 3-1 cells (**a**) vs. TPC-1 cells (**b**). Gene expression was analyzed by Real-time qPCR. The histogram represented normalized data with *GUSB* gene, and the results showed the average of three independent experiments. LY—LY294002; PD—PD98059; *—*p* < 0.05, **—*p* < 0.01 ***—*p* < 0.001 treated vs. quiescent cells; #—*p* < 0.05, ###—*p* < 0.001 cells with similar treatment.

**Figure 5 cells-11-00822-f005:**
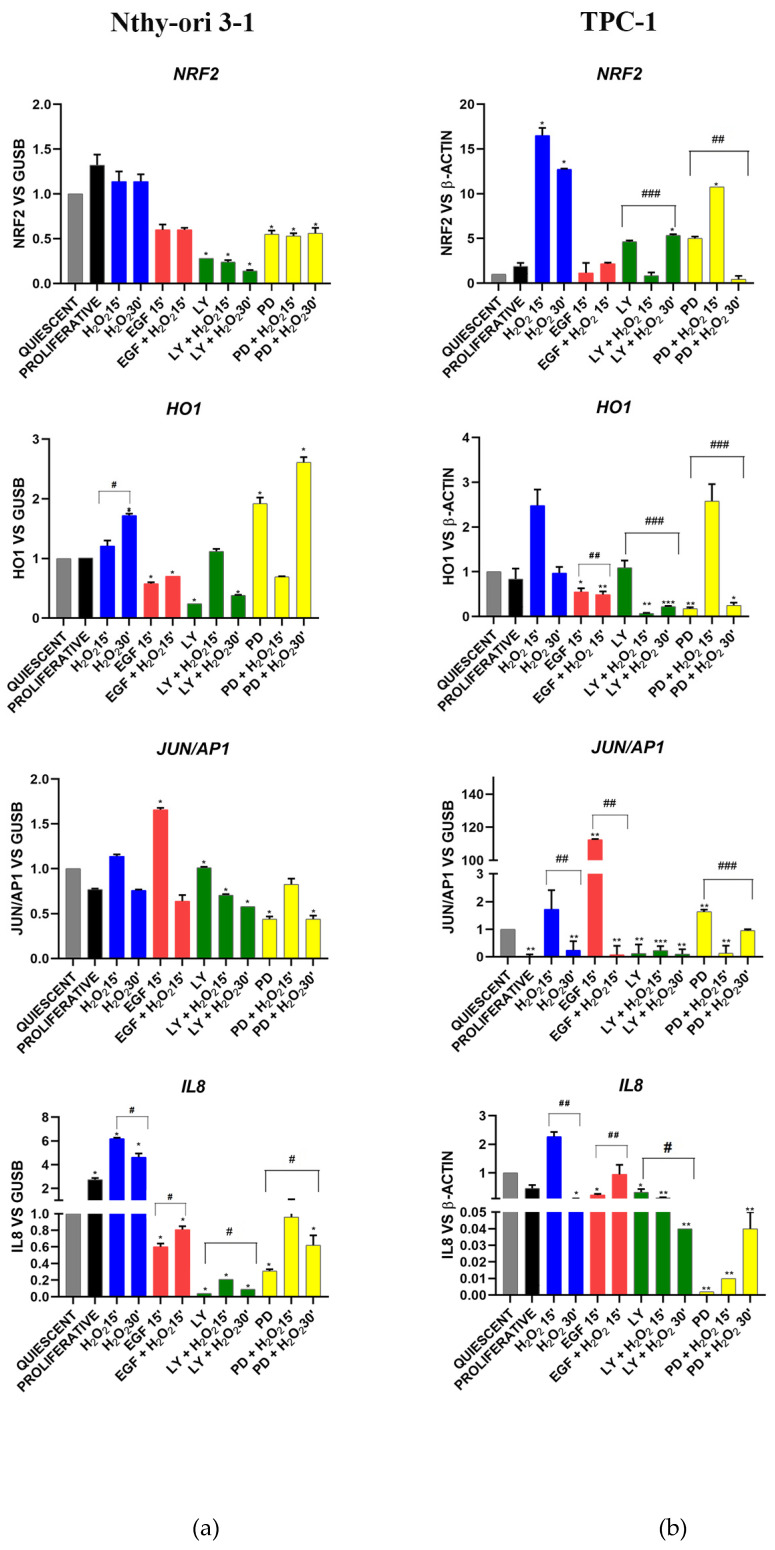
*NRF2* (nuclear factor erythroid 2–related factor 2), *HO1* (heme oxygenase 1), *JUN/AP1* (Jun proto-oncogene, *AP-1* transcription factor subunit) gene expression and *IL8* (*CXCL8*, interleukin 8) modulation under H_2_O_2_, EGF, LY and PD treatments alone and combined in Nthy-ori 3-1 cells (**a**) vs. TPC-1 cells (**b**). Gene expression was analyzed by real-time qPCR. The histogram represented normalized data with *GUSB* gene in Nthy-ori 3-1, while *NRF2* and *HO1* expression data were normalized with *β-Actin* in TPC-1 cells. The results showed the average of three independent experiments. LY—LY294002; PD—PD98059; *—*p* < 0.05; **—*p* < 0.01; ***—*p* < 0.001 treated vs. quiescent cells; #—*p* < 0.05, ##—*p* < 0.01, ###—*p* < 0.001 cells with similar treatment.

**Figure 6 cells-11-00822-f006:**
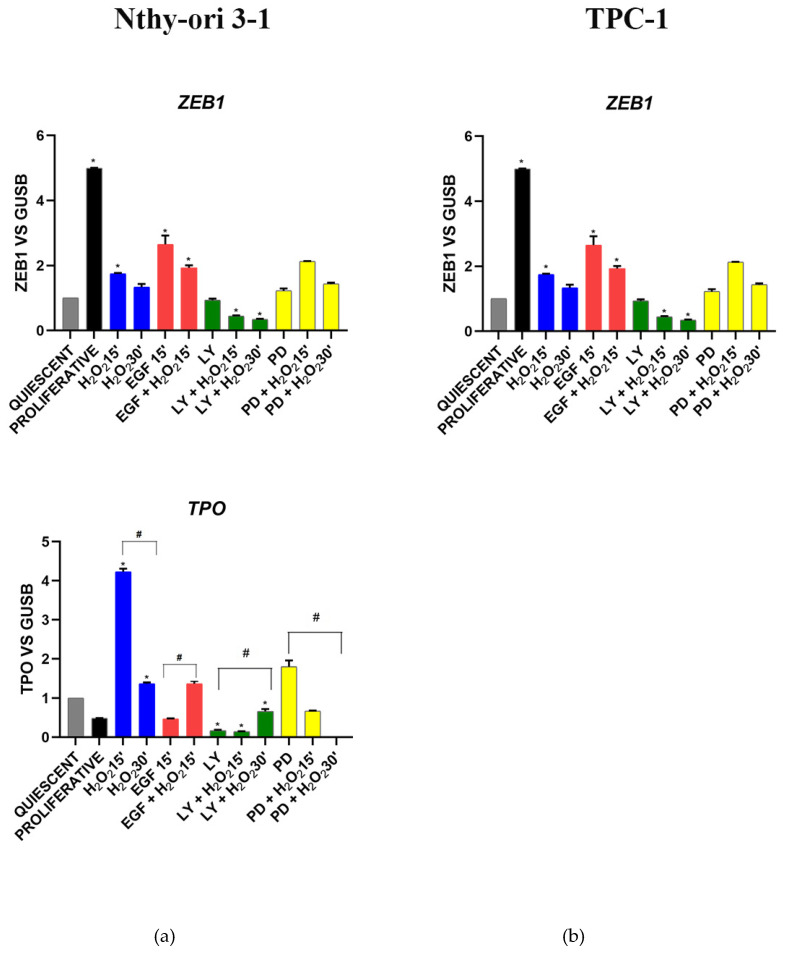
Gene expression modulation of *TPO* (thyroid peroxidase) and *ZEB-1* (zinc finger E-box binding homeobox 1) under H_2_O_2_, EGF, LY and PD treatments alone and combined. (**a**) Nthy-ori 3-1 cells, (**b**) TPC-1 cells. Gene expression was analyzed by real-time qPCR. The histogram represented normalized data with *GUSB* gene, and the results showed the average of three independent experiments. The expression of *TPO* was undetected in TPC-1 cells. LY—LY294002; PD—PD98059; *—*p*< 0.05 treated vs. quiescent cells; #—*p* < 0.05 cells with similar treatment.

**Figure 7 cells-11-00822-f007:**
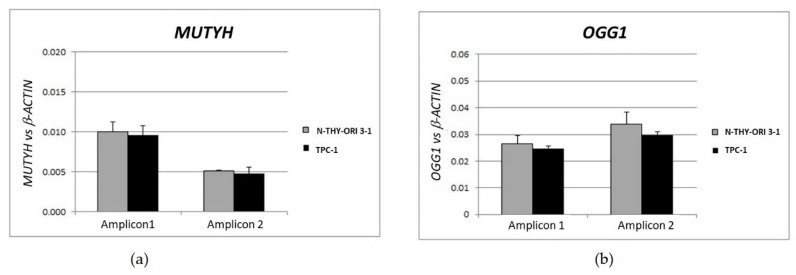
*MUTYH* and *OGG1* gene dosage. (**a**) Nthy-ori 3-1 cells, (**b**) TPC-1 cells. Gene dosage was performed by SYBR Green qRT-PCR. For both genes were selected two different genomic sequences. The samples were amplified in triplicate in three independent experiments. The gDNA amounts of the target genes were normalized by the ratio on median value of the *b-Actin* genomic reference gene from TPC-1 vs. Nthy-ori 3-1.

**Figure 8 cells-11-00822-f008:**
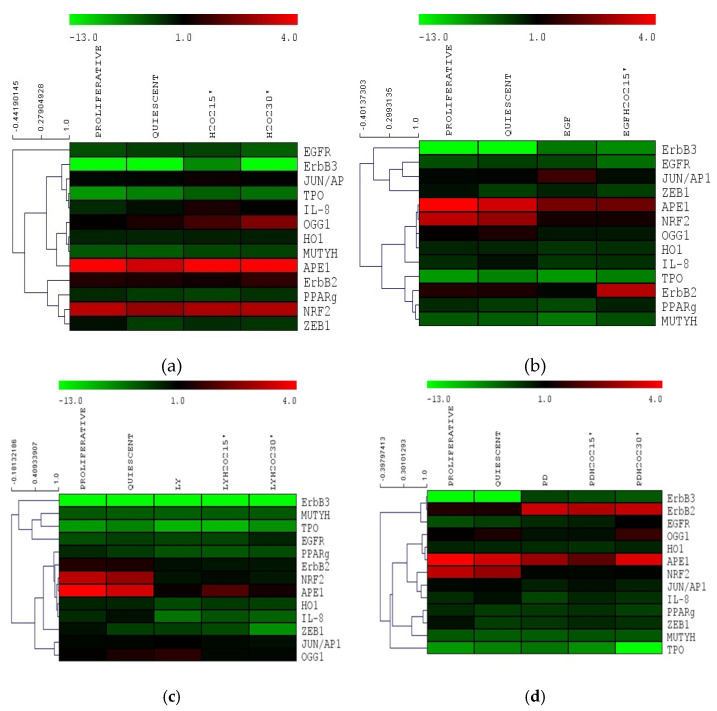
Gene expression cluster analysis by MeV4.9.0 in Nthy-ori 3-1 cells: (**a**) H_2_O_2_ treatments; (**b**) EGF treatments; (**c**) LY treatments; (**d**) PD treatments. The average linkage hierarchical clustering with Pearson correlation was used. The color scale at the top represents the log2 of every single gene expression value compared with housekeeping value ranging from −13 (green) to 4 (red). The trees presented here are the neighbor-joining trees based on gene expression variation in response to different treatments. LY—LY294002; PD—PD98059.

**Figure 9 cells-11-00822-f009:**
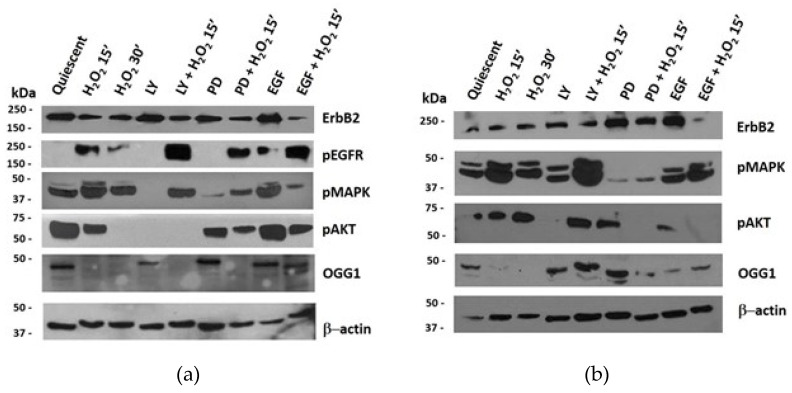
Protein expression and quantification in Nthy-ori 3-1 vs. TPC-1 cells. Nthy-ori 3-1 and TPC-1 cells were starved overnight and treated with, LY294002 (25 μM), PD98059 (50μM), EGF (50 ng/mL) alone and combined with H_2_O_2_ (10 mM). Western blotting analysis in Nthy-ori 3-1 cells (**a**) and in TPC-1 cells (**b**) determining the protein expression levels of p-EGFR, ErbB2, p-MAPK, p-AKT and OGG1. Results are representative of three independent experiments. Quiescent cells were treated with H_2_O_2_ and LY, PD, EGF alone or combined. LY—LY294002; PD—PD98059; kDa—protein molecular weight marker.

**Figure 10 cells-11-00822-f010:**
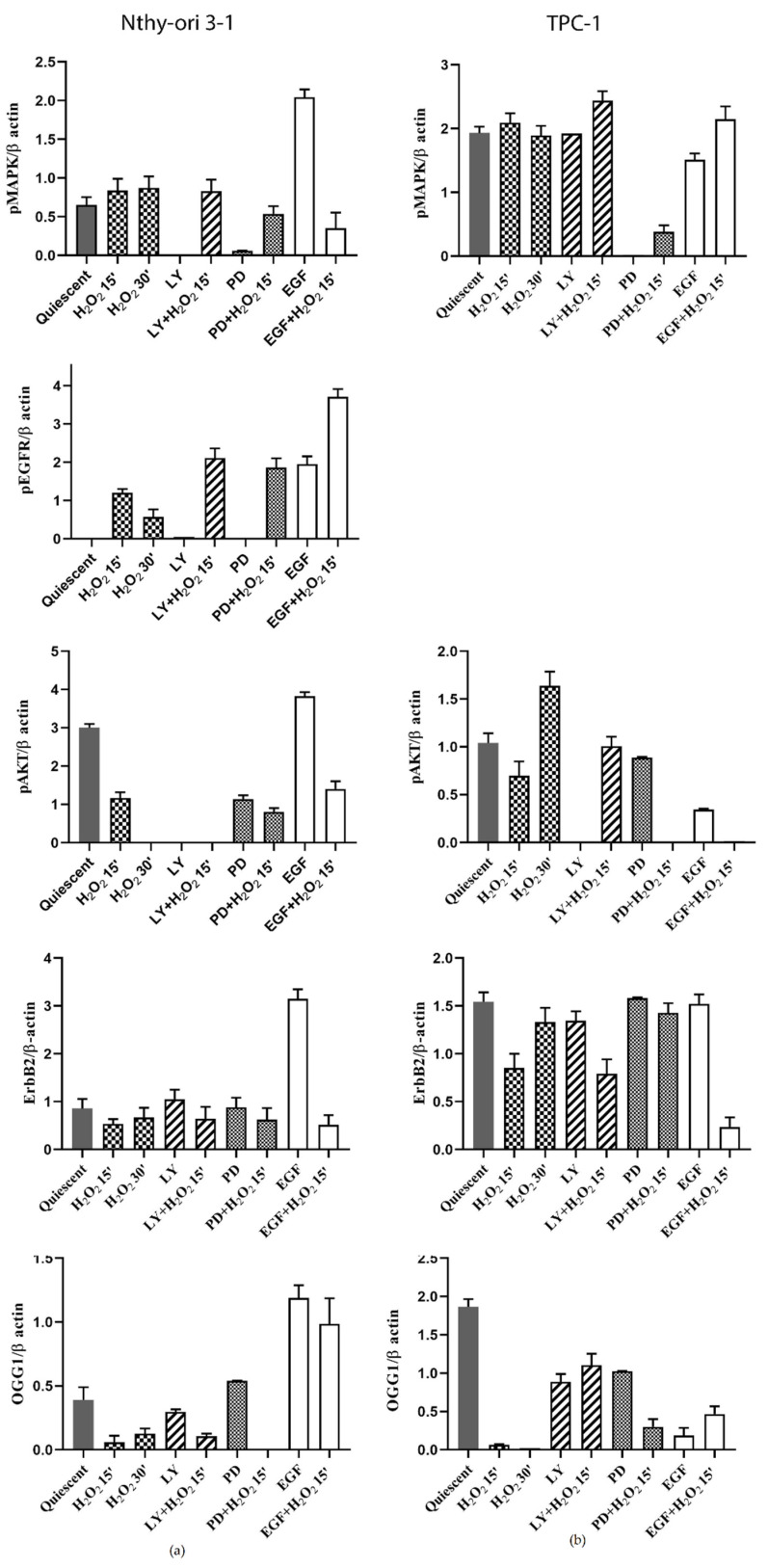
Protein expression quantification of ErbB pathway NThy-ori 3-1 vs. TPC-1 cells. Quantification of selected proteins (p-EGFR, ErbB2, p-MAPK, p-AKT) detected by WB in Nthy-ori3-1 and TPC-1 (Figure 9a,b). The expression of p-EGFR, was undetected in TPC-1 cells. The average expression levels of panel (**a**) and panel (**b**) were determined by densitometric analysis and calculated in relation to the β-Actin level. Quiescent cells were treated with H_2_O_2_ and LY, PD, EGF alone or combined. LY—LY294002; PD—PD98059.

## Data Availability

The datasets used and/or analyzed during the current study are available from the corresponding author upon reasonable request.

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
