# Peer review of "Emerging Role of Oxidative Stress on EGFR and OGG1-BER Cross-Regulation: Implications in Thyroid Physiopathology"

_cells, 2022, doi:10.3390/cells11050822_

Round 1

Reviewer 1 Report

This is an interesting study where the researchers demonstrated the cross- 
regulation between the ErbB and OGG1-BER pathways in human thyroid cells. They experimentally analyzed this regulation using a normal thyroid epithelial cell model and a papillary thyroid tumor model and this regulation was found to be dependent on both H2O2- and EGF-linked oxidative signals. Overall, the hypothesis was supported by the results and the data presentation was of good quality. However, to improve the manuscript, I would like to recommend the following:

  1. Please try to incorporate one more thyroid cancer cell line to better validate the outcome.
  2. Please restructure the sentence "In normal............TPO". (sentence 50-53)
  3. Please add "it" after "since" in sentence 67.
  4. Please restructure the sentence "Along............damage". (sentence 72-74)
  5. Please restructure the sentence "All............deviation". (sentence 185-186)
  6. Please restructure the sentence "H2O2............population". (sentence 215-216)
  7. Please include individual beta actin figure for each protein of interest to show equal loading of protein samples in Fig. 9.

Reviewer 2 Report

1) In materials and methods section, authors describe using MTS assay to measure "cell proliferation, viability and cytotoxicity" while in fact they conduct experiments on high density cells, following up maximum of 24h post treatment, which is not proliferation assay, only viability assay.

It should also be noted somewhere in the manuscript, that MTS assay is not a direct measurement of cell number/viability but their metabolic activity. Mitochondrial health and function is most likely affacted by oxidative stress, so additional measurement of cell viability would be useful to determine cell death. (Like PI/annexin V FACS stain)

2) there is no indication that the cell cycle experiments were repeated more than once. If they were not, these results should not be shown. If they were, a mean with error bars should be represented on the bar graphs.

3) The description of qPCR results is hard to follow, could be advantageous to split it into functional parts

4) There is no indication if Western Blot experiment was repeated. If it was, the quantification graphs should represent a mean and error bars. Description of figure 9 mentions panel c and d which are not there.

5) Shift in figures' numbering in text should be looked at (For example fig 9 is most likely described as fig 8 in discussion)

6) In Nthy-ori cells upon H2O2 treatment AKT activity decreases in all tested conditions, while in TPC-1 cells, AKT activity increases upon H2O2 treatment. That seems the most important signalling difference between the two models that should be addressed. How do authors explain the increase of AKT activity in TPC-1 cells after h2O2 exposure even in the presence of AKT inhibitor? But a decrease in the prescence of MAPK inhibitor and EGF? This unusual AKT activation is also followed by OGG1 highest protein level in LY+H2O2 condition, which is not reflected in mRNA levels. Did authors check the stability of OGG1 protein in these conditions? How could that be explained?

7) In discussion there is relatively a lot of focus on the role of PARP, while no results regarding this protein are shown.

Overall the article provides good insight into the effect of EGF signalling pathways on OGG driven repair systems, but the results and conclusions could be presented more clearly.
